# Mechanism Analysis of Discharge Energy in the Electrostatic-Field-Induced Electrolyte Jet Micro-EDM

**DOI:** 10.3390/mi14101919

**Published:** 2023-10-10

**Authors:** Yaou Zhang, Xiangjun Yang, Qiang Gao, Jian Wang, Wansheng Zhao

**Affiliations:** 1State Key Laboratory of Mechanical System and Vibration, Shanghai Jiao Tong University, Shanghai 200240, China; 2School of Mechanical Engineering, Shanghai Jiao Tong University, Shanghai 200240, China

**Keywords:** discharge energy, micro-EDM, E-Jet, surface charge density, electrolyte concentration

## Abstract

The discharge energy determines the machining resolution, minimum processable feature size, and surface roughness, which makes it a hot research topic in the microelectrical discharge machining (EDM) field. In this paper, a kind of novel discharge-energy-generation method in micro-EDM is investigated. In this method, the opposite induced charges on the electrolyte jet and workpiece serve as the source of the discharge energy. The operating mechanism of this discharge energy is revealed by analyzing the equivalent discharge circuit. The unique discharge current and voltage between the electrolyte jet and the workpiece are sampled and investigated. In contrast with the pulsating energy in conventional EDM, this study shows that the direct current (DC) voltage source can automatically generate a continuously periodical pulsating discharge in the electrostatic-field-induced electrolyte jet (E-Jet) EDM process. After further analyzing the electric signals in a single discharge process, it can be found that the interelectrode voltage experienced a continuous sharp electric breakdown, a nearly unchanging process, and a fast exponential recharging process. The discharge frequency increases as the electrolyte concentration and interelectrode voltage increase but decreases as the interelectrode distance increases. The discharge energy per pulse increases with the increasing interelectrode distance and electrolyte concentration but with the decreasing interelectrode voltage. Finally, the electrostatic-field-induced discharge-energy generation and change mechanisms are revealed, which provides a feasible method for micro-EDM with continuous tiny pulsed energy only using the DC power supply.

## 1. Introduction

In the micro-EDM process, the material is eroded in discrete units after each discharge between two electrodes. The amount of material removed per pulse is primarily determined by the discharge energy applied to the electrodes [1]. As a result, the discharge energy is highly related to the machining resolution, minimum processable feature size, and surface roughness [2], which has received much attention in the micro-EDM field.

In general, two kinds of fundamental pulsating discharge-energy-generation methods are preferred in micro-EDM [3], the resistance–capacitance type (RC) [4] and the transistor type [5]. The RC pulse power source depends on varying the capacitance values in the circuit to adjust the discharge energy. Therefore, the tiny pulsating energy can be obtained by reducing the capacitance designed in the RC circuit and the stray capacitance that exists between the electric feeders, tool electrode holder, and worktable, as well as between the tool electrode and the workpiece [6,7]. In fact, the designed capacitance in the circuit can be reduced to zero; however, the stray capacitance cannot be rooted out [8]. This means that the minimum discharge energy per pulse can be determined by the stray capacitance in the RC-type power supply.

On the other hand, a transistor-type controlled power supply relies on passively and transiently switching the transistor on and off to generate tiny bursts of pulsating energy. The biggest advantage of this transistor-type power supply is that it can supply the sufficiently high current. This means that the magnitude of the energy per pulse can only be obtained by decreasing the response time of the control-signal-generating circuit and the powering off/on time of the transistor [9]. However, even with the most advanced transistors, it is still difficult to reach the constant discharge duration with several nanoseconds. As a result, when it comes to microfabrication, the RC-type pulse generator is still the best solution [10].

Because of these facts, discharge-energy-generation methods have always been a hot topic among researchers. Many composite power supplies are proposed based on these two fundamental pulsating discharge-energy-generation theories to improve the machining performance [10,11]. The two-stage current-limiting resistance with a high open voltage and low discharge current is proposed to decrease the pulsating discharge energy to reduce the heat-affected zone and improve the surface finish [12]. Yan [13] suggested the transistor-controlled pulse generator with a high-power source circuit, a low-power source circuit, and an advanced logic controlling unit to reduce the surface finish below Ra0.247. The transistors with ultra-high-speed switching within two independent pulse generators have been presented to improve the quality of the machined surface [14]. Although progress has been made with these composite methods or advanced devices, the new fundamental theory has rarely been mentioned until Kunieda et al. [15,16] suggested an electrostatic induction feeding method to eliminate the stray capacitance influence. Zhang et al. [17] proposed an electrostatic-field-induced electrolyte jet (E-Jet) EDM method. Although this method depends on the electrostatic-field-induced theory, it is quite different from that suggested by Kunieda [15]. In this method [17], the tool electrode is the pulsating liquid E-Jet, which requires no tool electrode fabrication, no tool electrode wear, and no compensation. The energy comes from the discharge between the opposite induced charges on the jet and the workpiece directly, which means that there is no need to provide a dedicated pulsed power supply, and the drawbacks of pulsed power supplies in generating tiny amounts of discharge energy can all be avoided. These advantages make it a promising micro-EDM method in reality. In contrast to the conventional EDM, the energy comes from the direct current (DC) power supply, and the energy-generation process is coupled with the electrolyte jet tool electrode. To analyze this energy, we qualitatively study the generation mechanism and quantitatively assess the impact factors of this new energy-generation method.

This paper is organized as follows: in Section 2, the equivalent discharge circuit model is built to explain the principle of the electrostatic-field-induced energy-generation method; in Section 3, the experimental platform is established, and the experiment process and results are discussed; in Section 4, the current and voltage characteristics during the machining are further studied, and the factors affecting the discharge energy and frequency are analyzed; and in Section 5, the conclusions are drawn.

## 2. Materials and Methods

### 2.1. The Principle of E-Jet EDM

Figure 1 demonstrates the periodic E-Jet EDM process. Two electrodes, a flat-headed nozzle, and a workpiece are connected to two ends of the high DC power supply. Put the electrolyte into the nozzle. When the voltage is applied, the intense electric field emerges, resulting in the opposing charges redistributing inside the electrolyte and on the workpiece surface, which follows the Boltzmann distribution theory [18]. The combined effects of the electric field force, surface tension, and capillary effect can draw the electrolyte out from the nozzle due to the fluidity characteristics of the electrolyte. The liquid cone appears at the flat-headed nozzle and then develops gradually into the Taylor cone, as shown in Figure 1a. With the induced ions accumulating on the electrolyte surface, the rising electric field forces overcome the surface tension, resulting in the jet ejecting from the cone tip. This process is quite similar to the jet in electrospinning machining [19]. However, in the E-Jet EDM, the stretching of the jet shortens the distance from the workpiece surface and increases the electric field intensity, and the increasing electric field intensity grows greater than the breakdown electric field of the air, resulting in the electric discharge breakdown and eventually resulting in the formation of a plasma channel, as illustrated in Figure 1b. Following the discharge, the consumed induced charges lead to the decrease in the electric force, resulting in the jet retreating, which further leads to the electric field intensity decreasing and the discharge interruption. As a result, the cone quickly returns to its original shape, as seen in Figure 1c. With the increase in the induced charges gathering on the cone, the jet appears again. Then, the periodic E-Jet EDM process is formed. Along with the material removal, the balance is upset and moves the nozzle closer to the workpiece, and a new equilibrium can be built. The continuous, spontaneous, and periodical discharge process can be established [17].

The induced charges on both sides of the electrolyte cone and workpiece surfaces are the energy sources. The flowability of the electrolyte jet electrode and the DC power supply distinguish this discharge method from the conventional one and the electrostatic induction feeding method [15].

### 2.2. The Analysis of the Equivalent Circuit of the E-Jet EDM

During a discharge cycle in the E-Jet EDM process, the electrolyte at the nozzle outlet consecutively experiences the meniscus cone, the Taylor cone, the jet, and the withdrawal process. These states occur and shift instantaneously. The distance between the liquid cone tip and the workpiece surface is always in a dynamic change process. As a result, the workpiece and the jet (cone) tip can be treated as two substrates of the variable capacitor. The DC power supply, which may be regarded as an ideal constant-voltage source, is linked to the nozzle and the workpiece. Zhou [20] simulated the voltage-changing process with a charged capacity discharging through a resistor in the conventional EDM. A similar equivalent circuit model has been used in nanoelectron spray systems [19]. With a similar approach, the corresponding circuit of the E-Jet EDM process can be abstracted as Figure 2.

Figure 3 depicts the equivalent circuit of the E-Jet EDM process before, at, and after the discharge moment. To ensure the malleability of the charged liquid in the electric field and the tip effect, the charges aggregate towards the tip, and the electric field forces on the induced charges pull the liquid cone tip until it gradually begins approaching the workpiece, as shown in Figure 1a, and this can be treated as incremental capacitance. Figure 3a depicts the equivalent circuit of this stage before the discharge. As shown in Figure 1b, when the jet ejects from the cone tip, the electric field intensity between the jet tip and the workpiece surpasses the breakdown electric field, resulting in the discharge. Ignoring the plasma resistance, the circuit during this discharge process can be simplified by connecting two terminals of the power supply, as shown in Figure 3b. The discharge occurs in such a flash instant that the induced charges are swiftly consumed before the jet draws back, as seen in Figure 1c. As seen in Figure 3a, this drawing-back process automatically generated the deionization process in the EDM and created a decrescent capacitance.

## 3. Experimental Methods

### 3.1. Experimental Platform Setup

Figure 4 constructed the experimental platform to study the discharge energy and verify the discharge equivalent circuit during this E-Jet EDM. In contrast with the conventional EDM, the DC power source is adopted instead of the pulse power supply, which generates an intense electric field to draw the liquid out of the nozzle and induce enough charges on the liquid surface. Here, the commercial Spellman DC power supply is selected with a voltage range from 0 to 10 kV.

The silicon wafer placed on a conductive block is chosen as the workpiece. The sodium chloride electrolyte is kept inside the plastic nozzle, where the wire sticks inside to apply the voltage. The gap voltage is measured by using a Tektronix high-voltage probe (P6015A). The Tektronix current sensor (TCP0030A) is selected to measure the discharge current. The discharge current and voltage are sampled online by using a Tektronix oscilloscope (MDO3104 mixed-domain oscilloscope) at 1 GHz.

Although the discharge energy comes from the opposite charges on the workpiece and E-Jet surface, the valid way to study the energy is to identify the discharge current and discharge voltage because directly observing and calculating charges in the transient discharge process is almost impossible. On the other hand, the electric signals between the jet tip and the workpiece can only be measured indirectly. As a result, the voltage probe is linked to the wire inside the liquid and the workpiece to approximate the electric signals between the two ends of the discharge plasma.

### 3.2. Experimental Design

The NaCl electrolyte is chosen because of its strong electrical conductivity and ability to provide sufficient ions. Furthermore, it is safe, nontoxic, nonpolluting, inexpensive, etc. The silicon wafer is selected as the workpiece because of its mirror-like surface and because the tiny machined crater can be easily identified.

The electrolyte concentration, the nozzle-to-workpiece distance, and the voltage are three elements which are taken into account since they are highly related to the number of induced charges. The NaCl is bathed at a steady temperature of 90 °C for 3 h before it is thoroughly dissolved in the water. The electrolyte concentrations are 5 wt.%, 10 wt.%, and 15 wt.%. Move the micromotion platform in Figure 4b to adjust the distance between the nozzle and the workpiece. At the shortest point, the jet might directly touch the workpiece with no discharge but a water bridge in between. At the farthest point, a liquid jet with the longest period can reach the workpiece. These distances are often determined by the electrolyte concentration and the control voltage.

To eliminate the effect of discharge randomness on the single pulsating energy, ten experiments were carried out for each processing condition. During the data-processing procedure, remove the largest and smallest numbers, and the mean value of the remaining values is calculated and recorded.

## 4. Signal Acquisition and Analyses

### 4.1. Gap Voltage and Current Signals

Figure 5a shows a train of stable periodic pulsating current and voltage in the normal E-Jet EDM process. This process can confirm the fact that the discharge appears instantaneously. It has gone through the process of discharging (stage (i)) and recharging (stage (ii) and (iii)). The voltage is almost approaching zero in the transient process, and then there exists a vibration stage with a stable voltage, around 400 V for the mean (stage (i)). Then comes the roughly static voltage (stage (ii)) followed by a fast and nearly linear increase (stage (iii)) until the second discharge arrives. In the meantime, the current first vibrates and then remains zero until the next one comes, as depicted in Figure 5b.
(a)Voltage and current sampling experimental result(b)The discussion of the voltage and current signals

The pulsating current and voltage in E-Jet EDM differ significantly from the current–voltage characteristics in the electrospray process, in which the current rises in proportion to the voltage [21]. This procedure also differs from the traditional EDM procedure [4], in which the open voltage is 70–100 V and the discharge-sustaining gap voltage is about 0–20 V followed by the exponential recharging process [22]. However, in this method, the maintaining voltage is around 400 V while the open voltage is from 2.5 kV to 3.5 kV. When the plasma appears between the jet tip and the workpiece, the discharge current comes into being, and then an equivalent resistance of the plasma is generated [23]. At this moment, the deviation in the slim and soft jet leads to the plasma swaying in space. Furthermore, the vibrations of the voltage and current might be caused by the violent random collisions of the ions in the plasma channel. These might be the reasons why the vibrating is maintained in the voltage and current during the discharges.

In the meantime, the power supply (about 2.5 kV) is by far larger than that of the conventional EDM (approximately 70 V), resulting in a larger dividing voltage in the equivalent resistance of the plasma. Moreover, because it is impossible to directly measure the voltage between the soft E-Jet tip and workpiece, the voltage measured also includes that between the nozzle and E-Jet tip. As a result, at the E-Jet discharge moment, the larger maintained voltage appears.

The voltage in stage (ii) lies between the discharge-maintaining voltage and the open voltage. This might be caused by the fact that the discharge depleted numerous induced charges before the jet finished retracting. The induced charges on the jet surface are deficient and almost unchanged during this fallback process, and the equivalent variable capacity is decreasing. Because there is no current during this short interval, the gap voltage remains nearly constant.

After the jet finishes its retraction, the cone starts growing and the induced charges slowly accumulate on the jet surface again in stage (iii). As a result, the gap voltage begins to experience the same fast exponential recharging as that in the traditional RC power supply. At this point, the gap voltage gradually increases, then the cone is stretched, and the increasing capacitance is formed with more induced charges, as seen in Figure 3a.

This unique voltage and current is commensurate with the jet evolution process. In contrast to the solid tool electrode in conventional EDM [4,22], because of the appearance of the withdrawal process of the E-Jet tool electrode, there is a stage with a constant voltage generated.

### 4.2. Discharge Frequency Acquisition and Analyses


(a)Discharge Frequency Experimental Results


The discharge frequency is closely related to the material removal rate in the EDM process. The EDM utilized overlapping craters after each discharge to process the workpiece. As a result, the discharge frequency should be taken into account when addressing the discharge energy.

The electrolyte concentration, interelectrode distance, and applied voltage are explored. Three NaCl electrolyte concentrations were chosen: 5 wt.%, 10 wt.%, and 15 wt.%. The voltage ranges from 2.45 kV to 3.5 kV, and the nozzle–workpiece distance is kept at 0.55 mm.

Figure 6 depicts the effects of the applied voltages and electrolyte concentrations on the discharge frequency. The frequency increases with the growth in the voltage intensity, ranging from a few Hertz to 1000 Hz. The rising of the electrolyte concentration results in the increasing frequency.

Furthermore, fixing the interelectrode gap, there exists a start voltage. The initial voltage is 2.45 kV when the concentration is 15 wt.%. When the concentration is about 10 wt.%, the start voltage rises to 2.5 kV. Moreover, when the voltage exceeds 2.85 kV, the frequency rises fast with the increasing electrolyte concentrations.

Figure 7 provides the effect of the interelectrode distances on the discharge frequency. The electrolyte concentrations are set at 5 wt.%, 10 wt.%, and 15 wt.%, and the interelectrode distances are reduced from 0.65 mm to 0.3 mm at 0.05 mm. The interelectrode voltage is fixed at 2.6 kV.

The interelectrode distance affects the frequency in E-Jet EDM. The frequency slightly decreases when the interelectrode distance increases from 0.45 mm to 0.6 mm. The frequency decreases dramatically when the interelectrode gap increases from 0.3 mm to 0.45 mm. Moreover, when the interelectrode distance decreases further, the electrolyte might touch the workpiece, resulting in a water bridge instead of the discharge. On the other hand, when the interelectrode distance grows to 0.65 mm, the frequency drops to several Hz. The electrolyte concentration also affects the discharge start distance. When it is 15 wt.%, the start distance is 0.65 mm. When it reduces 5 wt.%, the start distance decreases to 0.6 mm.
(b)The discussion of the discharge frequency

At a given range, the frequency decreases as the interelectrode distance grows. The increase in the interelectrode distance results in a decreasing equivalent capacity, leading to a decline in the induced charge quantity and density and resulting in a decrease in the recovery speed of the electrolyte cone and electrolyte jet. When the interelectrode distance exceeds the upper critical distance, the discharge seldom appears, and the frequency is considered minuscule. In this case, the induced charges are insufficient to provide enough electric field force to overcome the surface tension [21]. On the other hand, the frequency increases dramatically with the decreasing interelectrode distance. This is because the decreasing interelectrode gap increases the equivalent capacity, resulting in the growing electric field intensity, gathering speed, and number of induced charges on the jet surface. At last, a pulse with a short duration and interval can be achieved. Furthermore, when the interelectrode distance drops to the smallest value, the water bridge appears and the frequency changes to zero.

The frequency increases with the increasing applied voltage. Initially, the increasing voltage causes the growing electric field intensity. The increasing voltage leads to the growing-induced charges on the cone, resulting in the boosting repulsion force. These findings are consistent with those of Wei [24], who found that the increase in the applied voltage is correlated with the decrease in cone height. Furthermore, the increasing electric field intensity enhances the gathering speed of the induced charges on the reduced-height cone, resulting in a growing recharging speed. Moreover, when the applied voltage is too small, the electric field can only induce a small charge density on the jet surface. The charges are unable to provide enough electric field force to overcome the liquid surface tension. As a result, there exists the start voltage. Furthermore, the frequency increases rapidly with the growing applied voltage. It seems possible that the rise in the electric field strength and the growth of the number of ions in the electrolyte concentration result in a significant increase in the density and gathering speed of the induced charges on the electrolyte jet surface. These reasons shorten the time that the electric forces spend growing, rolling back, and recovering and increase the discharge frequency in the end.

On the other hand, the E-Jet is dragged by the electric field force without any pumps, and the large voltage can result in a more violent discharge, leading to a large retreated thrust. This force can make the liquid retreat inside the nozzle, leading to the interruption of discharges. Moreover, the large voltage can also ionize droplets [21], resulting in no E-Jet. As a result, there is an upper limit value for the voltage. In accordance with the present results, Borra [25] demonstrated that the formation speed of electrically charged droplets grows with increasing voltage. In his study, the workpiece and nozzle are placed vertically, and gravity has a greater impact on the drop than the electric field force. However, in the E-Jet EDM process, they are arranged horizontally, and the periodic jet is mainly controlled by electric force. Nonetheless, the analysis methods are comparable.

As seen in Figure 6 and Figure 7, the frequency increases as the electrolyte concentration increases. The increase in the concentration results in an increase in the number of free ions in the electrolyte, which leads to the growing charge density on the surface [24]. Although the growth of the concentration also increases the surface tension [26], the increasing free ions enhance the density and gathering speed of the charges on the jet surface, leading to electric forces growing significantly faster than the surface tension. As a result, the discharge time and recharge time in each E-Jet discharge process are reduced. These are possible explanations for the frequency growth with the increasing electrolyte concentration.

### 4.3. Discharge Energy Experimental Results and Analyses


(a)Discharge energy experimental results


The discharge of the induced charges between the jet surface and the workpiece surface generates the energy. According to the traditional EDM theory, the discharge energy per pulse can be stated as [27]
(1)$$E_{e}=\int_{0}^{t_{e}}u_{e}(t)\cdot i_{e}(t)\cdot dt$$

*u_e_* is the gap voltage, $i_{e}$ is the discharge current, and $t_{e}$ is the discharge maintenance time.

It is difficult to obtain the mathematical model to employ Equation (1) because the current and voltage are sampled online [28]. The accurate method to calculate the discharge energy can be expressed as
(2)$$E_{e}=U_{e}\cdot I_{e}\cdot t_{i}$$

Here, $U_{e}$ is the equivalent voltage, $I_{e}$ is the equivalent current, and $t_{i}$ is the discharge duration. According to Equations (1) and (2), the discharge energy is determined by the discharge voltage, discharge current, and pulse duration.

Figure 8a depicts the original voltage and current waveforms sampled online with a frequency of 5000 Hz. The discharge signals are filtered by using the 6th-order low-pass Butterworth filter with a cutoff frequency of 5 Hz, shown in Figure 8b.

Based on experience in electrospraying and EDM, the electric field and electrolyte concentration might affect the discharge energy. Faraji [29] presented that the conductivity and solution concentration play important roles in the formation of Taylor cones in the electrospraying process. As a result, when studying the discharge energy in E-Jet EDM, the electric field and electrolyte were investigated.

Figure 9 shows that the discharge energy per pulse reduces with the increasing interelectrode voltage but increases with the growing electrolyte concentration and the interelectrode distance. When the concentration is 5 wt.%, the discharge energy is roughly proportional to the interelectrode distance, as shown in Figure 9a. When the concentration grows to 10 wt.%, the discharge energy in Figure 9b changes more rapidly with the increasing interelectrode distance. However, when the concentration increases to 15 wt.%, as the interelectrode distance grows, the increased rate of the discharge energy slows down, as shown in Figure 9c. From Figure 9, it can also be found that the effective discharge distance of the interelectrode decreases with the increasing discharge concentration.
(b)Discharge energy experimental analyses

When the electric field intensity exceeds 1 × 10^6^ V/m, the discharge can be generated in the air [30]. Figure 9 presents that the discharge energy increases with the growing interelectrode distance. This finding was unexpected and suggests that the decrease in the electric field intensity results in increasing the discharge energy per pulse. A possible explanation for this might be that the jet must be slowly dragged further from the nozzle to make the electric field intensity between the jet tip and the workpiece exceed the breakdown electric field. At this time, the jet is slim and the discharge frequency increases. This is consistent with the conclusion that the discharge frequency decreases with the increasing interelectrode distance discussed in Section 4.2 and the jet shape presented in [24]. According to the jet-formation principle [23], the electric forces on the jet should outweigh the surface tension. As a result, more charges should be induced to generate higher drag forces in the decreasing electric field, which is caused by the growing interelectrode distance. Because the opposite induced charges are the source of the discharge energy, the increasing charges on the jet under the increasing interelectrode distance results in the increase in the discharge energy per pulse.

Figure 9 shows that the discharge energy per pulse diminishes with the increasing interelectrode voltage. This might be caused by the fact that the electric field intensity increases with the increasing interelectrode voltage, and smaller induced charges on the jet surface can generate enough electric field force to overcome the surface tension. Furthermore, as the applied voltage increases, the cone length decreases, and the electrolyte diameter grows. These findings are consistent with the behaviors reported during the electrolyte spray technique [31]. As a result, the discharge energy per pulse increases as the interelectrode voltage decreases.

When the concentration is low, the induced charges are spread more uniformly. This might be the reason that the discharge energy almost grows linearly with the increasing interelectrode distance, as shown in Figure 9a. However, with the increasing concentrations, more charges are induced on the jet surface, resulting in the increase in the inhomogeneity of the charge distribution on the jet surface. This causes an increase in the stochasticity of the discharge. These might be the reasons that the discharge energy has a nonlinear relationship with the interelectrode distance, which is shown in Figure 9b,c.

From Figure 9, it can be found that the discharge energy increases with the increasing concentrations at a certain interelectrode distance. The surface tension can be obtained by integrating the Gibbs absorption isotherm equation, which shows that the surface tension of electrolyte NaCl increases as the concentration increases [26]. As a result, the electric field force should increase to overcome the increasing surface tension to produce the jet. The increase in the electric field force depends on the increase in the induced charges. Consequently, the induced charges on the jet surface increase with the increasing electrolyte concentrations.

Moreover, it can be found in Figure 9 that the increasing concentration might result in the reduced effective discharge distance. This is because the increasing concentration results in the increasing surface charge density, which leads to an increase in impulsive forces between like charges. As a result, the higher concentrations might reduce the electrolyte jet length, resulting in the loss of the discharge ability between the E-Jet tip and the workpiece. This effect is consistent with the electrospray cone jet technique [19].

According to the above discussions, the discharge energy can be regulated by modifying the electric field strength, i.e., adjusting the distances and voltages and adjusting the electrolyte concentrations.

### 4.4. The Machining Experiments

A single-point machining experiment with the electrolyte NaCl 5 wt.% is performed to validate the method’s processing capacity. Because of its semiconducting properties, polished silicon is chosen as the workpiece, and the machined crater on the mirrored surface of silicon may be captured before using the microscope.

The interelectrode distance during the discharge process is 0.62 mm. The applied interelectrode voltage is 2.8 kV. A white point can be seen on the workpiece surface after 5 min of processing with the naked eye. The crater can be identified by using a Zeiss Smartzoom 5 microscope after being cleaned for 20 min by using ultrasonic methods and dried. The machined crater is shown in Figure 10. It can be found that a pit with a diameter of 50 μm and a depth of 18 μm can be machined, and the machined crater structure is a typical EDM morphology.

## 5. Conclusions

This paper focuses on the equivalent circuit of the E-Jet EDM process, the influencing factors, and the influence laws on the discharge energy. It was revealed that the interelectrode distance, interelectrode voltage, and electrolyte concentration all have significant effects on the discharge energy. The following conclusions can be reached:

(1). The experimental results showed that during the E-Jet EDM process, the stable periodic discharge voltage and current can be generated automatically under a high DC power supply. The interelectrode voltage experienced a breakdown, a nearly unchanging process, followed by an exponential recharging process.

(2). Compared to the conventional EDM process, the interelectrode voltage in the E-Jet EDM process experienced a discharge maintenance voltage of roughly 400 V.

(3). The discharge frequency increases with the increasing concentration and interelectrode voltage but decreases with an increasing interelectrode distance. The frequency increases with the growth in the voltage, ranging from a few Hertz to 1000 Hz. The frequency slightly decreases when the interelectrode distance increases from 0.45 mm to 0.6 mm and dramatically decreases when the interelectrode gap increases from 0.3 mm to 0.45 mm. The rising electrolyte concentration results in the increasing frequency.

(4). The discharge energy per pulse increases with the increasing nozzle-to-workpiece distance and electrolyte concentration but decreases with the increasing interelectrode voltage. When the concentration is 5 wt.%, the discharge energy is roughly proportional to the interelectrode distance. When it grows to 10 wt.%, the discharge energy changes rapidly with an increasing interelectrode distance. However, when the concentration increases to 15 wt.%, as the interelectrode distance grows, the increased rate of the discharge energy slows down.

## Figures and Tables

**Figure 1 micromachines-14-01919-f001:**
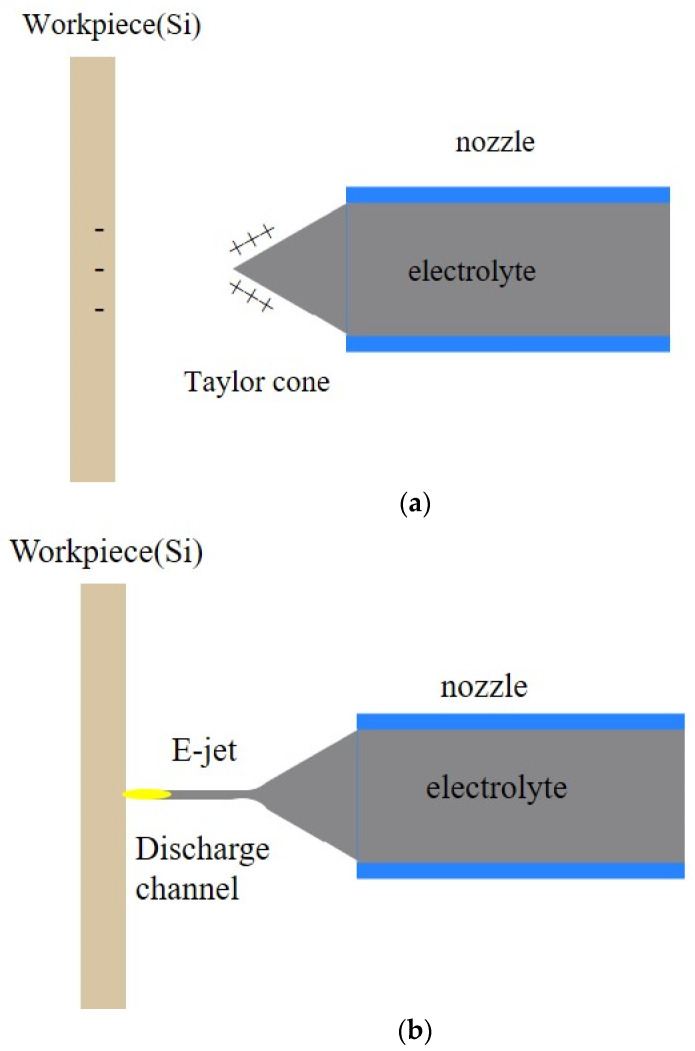
The single discharge process in the E-Jet EDM. (**a**) the preparation of the E-jet tool electrode, (**b**) the discharge of the E-jet process, (**c**) the disruption and rollback of the E-jet process.

**Figure 2 micromachines-14-01919-f002:**
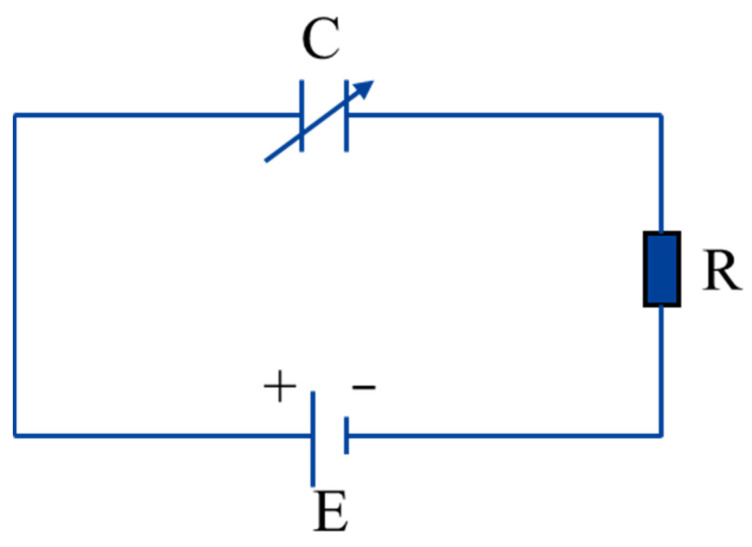
The equivalent circuit of the E-Jet EDM.

**Figure 3 micromachines-14-01919-f003:**
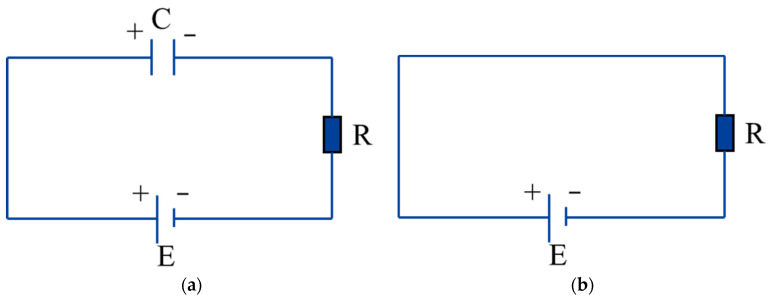
The equivalent circuit of the E-Jet EDM (**a**) before and after the discharge moment and (**b**) at the discharge moment.

**Figure 4 micromachines-14-01919-f004:**
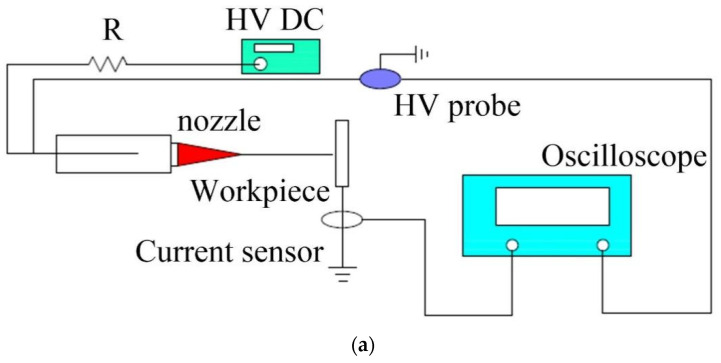
The experimental platform of the E-Jet EDM: (**a**) schematic diagram, (**b**) hardware platform.

**Figure 5 micromachines-14-01919-f005:**
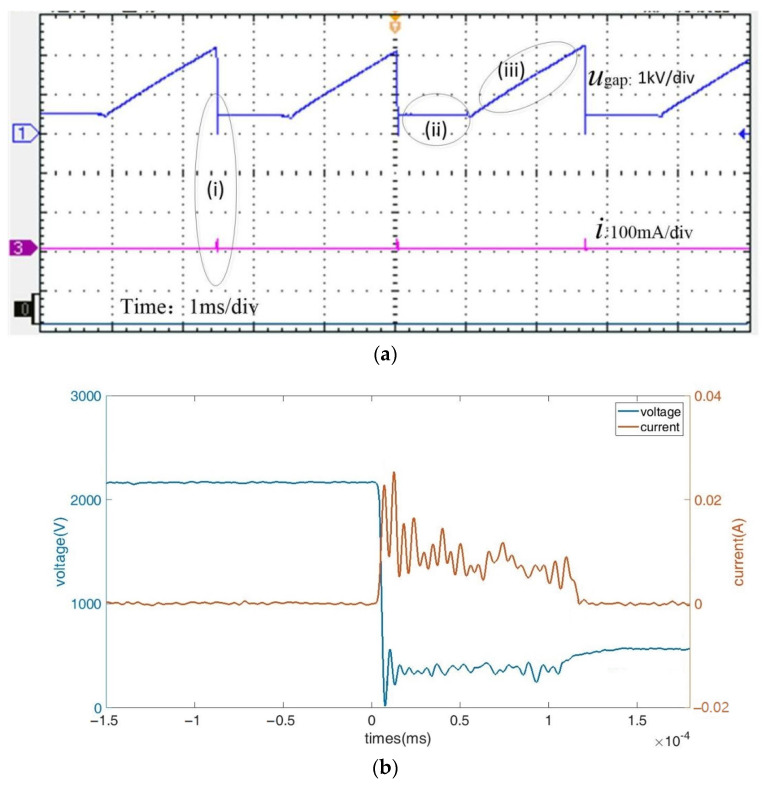
(**a**) A train of pulsating current and voltage; (**b**) a single pulsating current and voltage in the E-Jet EDM process.

**Figure 6 micromachines-14-01919-f006:**
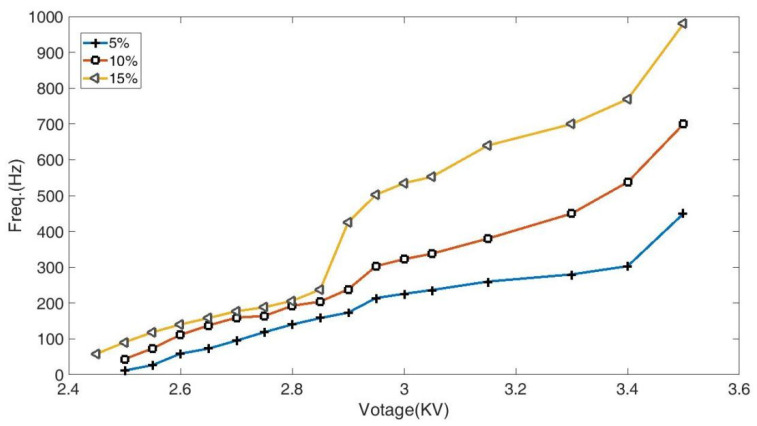
The frequency varying with the applied voltages and concentrations with the fixed interelectrode distance.

**Figure 7 micromachines-14-01919-f007:**
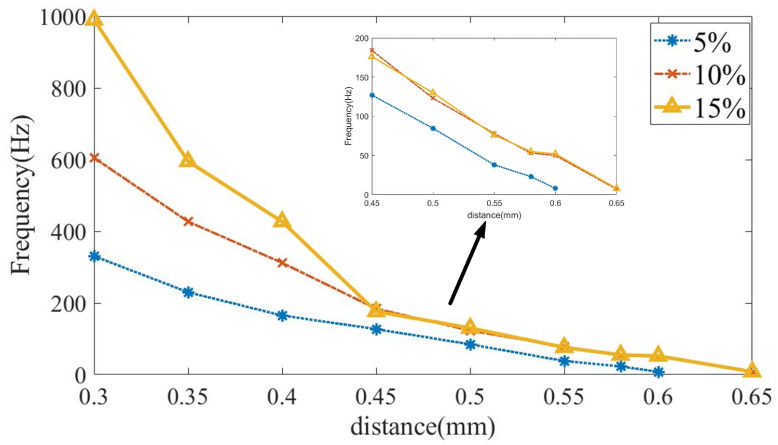
The interelectrode distance influence on the discharge frequency with 2.6 kV interelectrode voltage.

**Figure 8 micromachines-14-01919-f008:**
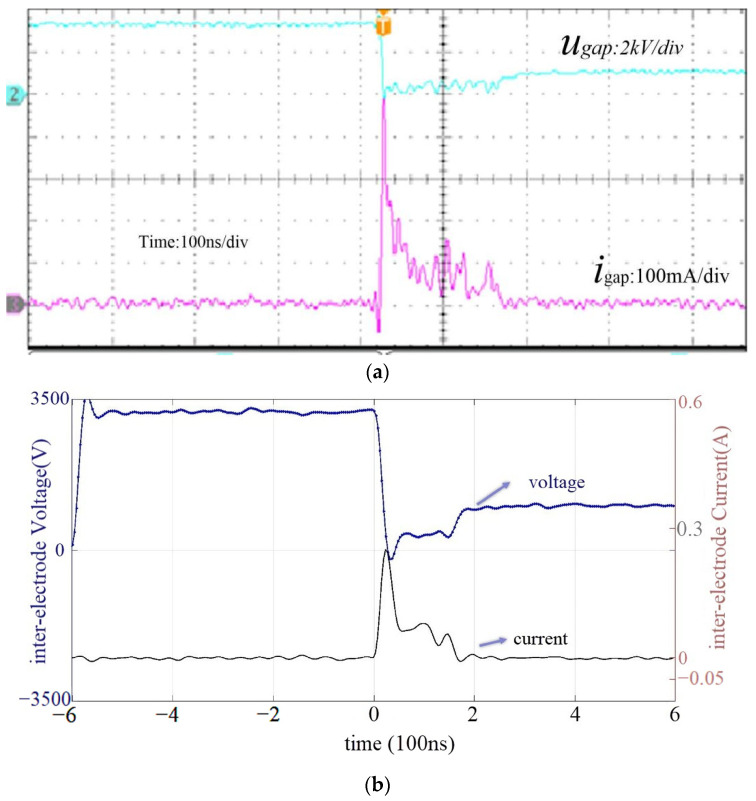
(**a**) The voltage and current originally sampled by the microscope. (**b**) The filtered voltage and current during a single E-Jet EDM process.

**Figure 9 micromachines-14-01919-f009:**
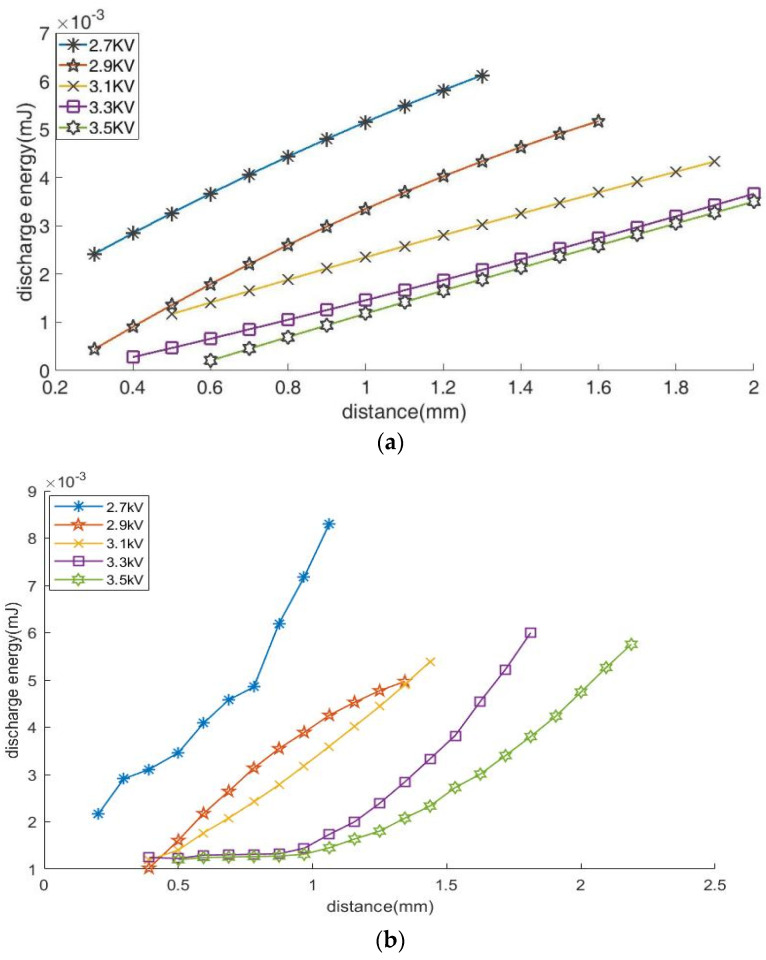
Discharge energy variation with different interelectrode distances and at different voltages and different concentrations: (**a**) 5 wt. %, (**b**)10 wt. %, and (**c**) 15 wt.%.

**Figure 10 micromachines-14-01919-f010:**
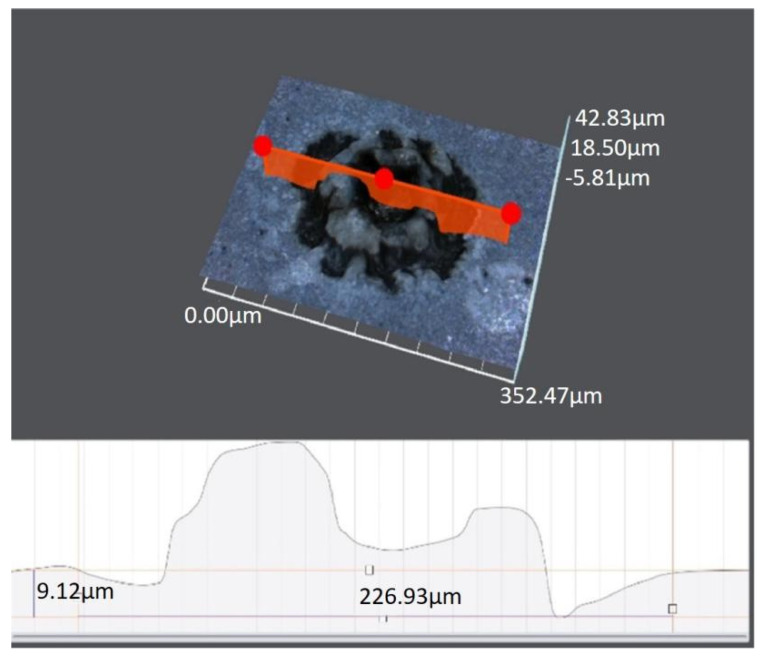
The etched crater morphology of E-Jet EDM.

## Data Availability

Not applicable.

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
