# Peer review of "Mechanism Analysis of Discharge Energy in the Electrostatic-Field-Induced Electrolyte Jet Micro-EDM"

_micromachines, 2023, doi:10.3390/mi14101919_

Round 1

Reviewer 1 Report

In this study, the process mechanisms of the E-Jet EDM depending on various process factors were investigated and discussed well. The outcomes seem beneficial to readers, so the manuscript is considered acceptable for publication. Before acceptance, on p.14 line2, “10e6V” seems to be miswritten. Please correct it.

Minor editing of English language is required.

Reviewer 2 Report

From my point of view, the work is fine. I have no special comments.

Please review the bibliography and the Conclusions (which need to be detailed more ! )

Reviewer 3 Report

This paper focuses on the equivalent circuit of the E-Jet EDM process, as well as the influencing factors and influence laws of the discharge energy. Some questions are proposed as follows:

(1)How the discharge channel formed while applying a high DC voltage should be explained further in section 2.1.

(2) The liquid cone tip was formed under the particular conditions should be added to the manuscript. Could the cone angle or height be controlled?

(3) the volume of electrolyte may be decreased with discharge times, how to guarantee the steady cone tip geometry?

(4) The unique advantages of the discharging method for micromachining should be further discussed.

(5) The processing results of metal using the proposed method may be added.

Minor editing of English language required.
